# Catalyst-free carbosilylation of alkenes using silyl boronates and organic fluorides via selective C-F bond activation

Jun Zhou [1], Bingyao Jiang[2], Yamato Fujihira[2], Zhengyu Zhao[1], Takanori Imai[2] & Norio Shibata [1,2,3] ✉

A regioselective carbosilylation of alkenes has emerged as a powerful strategy to access molecules with functionalized silylated alkanes, by incorporating silyl and carbon groups across an alkene double bond. However, to the best of our knowledge, organic fluorides have never been used in this protocol. Here we disclose the catalyst-free carbosilylation of alkenes using silyl boronates and organic fluorides mediated by $^t$BuOK. The main feature of this transformation is the selective activation of the C-F bond of an organic fluoride by the silyl boronate without undergoing potential side-reactions involving C-O, C-Cl, heteroaryl-CH, and even $CF_3$ groups. Various silylated alkanes with tertiary or quaternary carbon centers that have aromatic, hetero-aromatic, and/or aliphatic groups at the $\beta$-position are synthesized in a single step from substituted or non-substituted aryl alkenes. An intramolecular variant of this carbosilylation is also achieved via the reaction of a fluoroarene with a ω-alkenyl side chain and a silyl boronate.

[1] Department of Nanopharmaceutical Sciences, Nagoya Institute of Technology, Gokiso, Showa-ku, Nagoya, Japan. [2] Department of Life Science and Applied Chemistry, Nagoya Institute of Technology, Gokiso, Showa-ku, Nagoya, Japan. [3] Institute of Advanced Fluorine-Containing Materials, Zhejiang Normal University, Jinhua, China. ✉email: nozshiba@nitech.ac.jp

The use of two independent reactants for the transition-metal-catalyzed difunctionalization of alkenes has emerged as a powerful strategy to access molecules with multiple functional groups, by incorporating two new functional groups across an alkene double bond[1–6]. One of the two coupling partners commonly used in this context are activated halogenated arenes or alkanes, or an equivalent thereof (Ar-X or Alkyl-X; X = e.g., I, Br, Cl, OTf, or OMs). Reactions using these molecules generally proceed via oxidative addition and/or radical processes[7–15]. In contrast, fluorinated compounds such as aryl or alkyl fluorides have not yet been used as a counterpart for the difunctionalization of alkenes due to the high bond-dissociation energy of the carbon–fluorine (C–F) bond (Fig. 1a).

In recent years, a variety of organofluorine compounds has become ubiquitous due to the rapid establishment of technologies for their synthesis[16–18]. There are more than 340 registered fluoro-pharmaceuticals[19] with complex structures and more than 425 registered fluoro-agrochemicals[20]. It has been estimated that there are more than 8,800,000 commercially available fluoroarenes (SciFinder®). Thus, the functionalization of organic fluorides via the cleavage/activation of C–F bonds in organic fluorides is a dynamic and emerging area of research that will expand the utility of organofluorine molecules[21–24]. However, the chemical transformation of fluoro-organic compounds via C–F bond cleavage under very mild conditions is still rare. For example, aryl fluorides have rarely been functionalized due to the difficulty of oxidative addition of low-valent transition metals to $C(sp^2)$-F bonds, whereas the field of $C(sp^2)$-F cleavage using silyl cooperativity and/or transition-metal catalysis has also benefited from a wide breadth of research, especially in the last decade[24–30]. Activation of $C(sp^3)$-F bonds in aliphatic fluorocarbons is mostly limited to reactive benzyl or allylic fluorides where p-block-based Lewis acids with high fluoride affinity are usually required[31–33]. Thus, the activation of inert C–F bonds in aromatic and aliphatic organic fluorides under mild conditions remains highly challenging.

To realize the difunctionalization of alkenes using organic fluorides as one of the coupling partners, we decided to investigate the carbosilylation of alkenes[34–49]. This reaction provides functionalized organosilicon compounds directly from simple alkenes, compounds that are much sought-after materials in organic synthesis[50,51], electronics[52,53], photonics[54], and drug discovery[55]. Despite previous successes in the hydrosilylation of alkenes[56,57], research on the carbosilylation of alkenes is still in its infancy[34–49]. One of the early milestones in this area is the work by Terao and colleagues[37]. They disclosed a titanocene-catalyzed carbosilylation of alkenes using alkyl bromides and chlorides with chlorosilanes mediated by the Grignard reagent. For recent examples, Engle and colleagues[42] reported a Pd-catalyzed arylsilylation of alkenes using dimethylphenyl silyl boronate PhMe₂-SiBPin and aryl triflates. Zhang and Hu[49] demonstrated an arylsilylation of acrylates with tri(trimethylsilyl)silane and aryl bromides using a nickel/photoredox catalyst. Although several achievements have been reported in carbosilylation reactions of alkenes under metal-catalyzed and photo-induced conditions, the present methods are still limited by the substrate scope. Besides, the aryl and alkyl fluorides have never been used for the three-component carbosilylation of alkenes. Our group has been engaged in the development of efficient methods for the selective functionalization of $C(sp^2)$-F and $C(sp^3)$-F bonds under mild conditions both with[58] or without[58–61] the use of transition metals. In this context, our recent report on the defluorosilylation of organic fluorides using silyl boronates and with or without a Ni catalyst[58] inspired us to undertake a much more challenging research topic: the difunctionalization of alkenes with aryl or alkyl fluorides in the presence of silyl boronates via C–F bond cleavage (Fig. 1b).

Herein, we disclose a protocol consisting of the catalyst-free defluorinative carbosilylation of alkenes with silyl boronates and fluorinated compounds with an inert $C(sp^3)$-F or $C(sp^2)$-F bond. A wide variety of aryl fluorides and alkyl fluorides are smoothly incorporated into the alkenes via the cleavage of a C–F bond in the presence of silyl boronates to provide β-functionalized silyl alkanes in good to excellent yield, without any help of transition-metal catalysis nor photoredox system. The alkene, aryl fluoride, and alkyl fluoride substrate scope tolerated by this reaction is extensive. Unsubstituted styrene derivatives as well as substituted and conjugated aryl alkenes react smoothly with a great variety of aryl or alkyl fluorides and silyl boronates. This allows access to a library of silyl compounds that regioselectively incorporate aryl, heteroaryl, and/or alkyl units at the β-carbon of the silyl alkanes. The reaction proceeds with high regio- and chemoselectivity. Aryl and conjugated alkenes are reactive, whereas non-aryl alkenes are entirely unreactive. The potentially cleavable C-O bond of ethers[62], C-Cl bonds[63], and the $C(sp^2)$-H bond of hetero-aromatic compounds[64,65] are well-tolerated. Most significantly, the $C(sp^3)$-F bond of the trifluoromethyl (CF₃) group[66] remains intact. An intramolecular carbosilylation via the cleavage and coupling of a C–F bond was also achieved. Three-component coupling reactions involving drug derivatives that contain a fluoride moiety were also demonstrated to prove the utility of this transformation in the drug discovery process. The reaction should proceed through a cascade radical process initiated by single-electron transfer, which was supported by the experimental studies.

## Results and discussion

**Optimization of the reaction conditions.** We first investigated the reaction of 4-fluorobiphenyl (**1a**) with styrene (**3a**) in the presence of the silyl boronate Et₃SiBpin and a variety of catalysts (Table 1, also see Supplementary Tables S1–S5 for more details).

**Fig. 1 Difunctionalization of alkenes. a** General figure for the difunctionalization of alkenes. **b** Carbosilylation of alkenes with silyl boronates and organic fluorides.

**Table 1 Optimization of the reaction conditions.**[a]

| Entry | Ni(COD)$_2$ (mol%) | Et$_3$SiBpin | Base (equiv) | Cyclohexane/THF | Yield (%)[b] |
|---|---|---|---|---|---|
| 1 | 10 | 1.5 | KOtBu (2.5) | 1/2 | 32 |
| 2 | 10 | 1.5 | KOtBu (3.5) | 1/2 | 38 |
| 3 | 10 | 1.5 | KOtBu (4) | 1/2 | 52 |
| 4 | 10 | 2 | KOtBu (4) | 1/2 | 81 |
| 5 | 10 | 2 | KOtBu (4) | 8/1 | 98 (95) |
| 6 | 1 | 2 | KOtBu (4) | 8/1 | 99 (94) |
| 7 | 1 | 2 | NaOtBu (4) | 8/1 | 15 |
| 8 | 1 | 2 | 0 | 8/1 | 0 |
| 9 | 1 | 2 | LiOtBu, KOMe or KHMDS (4) | 8/1 | 0 |
| 10[c] | 1 | 2 | KOtBu (4) | 8/1 | 45 |
| 11 | 0 | 2 | KOtBu (4) | 8/1 | 94 (91) |

[a]Unless otherwise noted, the reaction was carried out using **1a** (0.1 mmol), **3a** (0.2 mmol), Et$_3$SiBpin, Ni(COD)$_2$, and a base in cyclohexane/THF (0.75 mL) at rt for 2.5 h.
[b]Yields were determined by $^1$H NMR and $^{19}$F NMR analysis of the crude reaction mixture using 3-fluoropyridine as the internal standard.
[c]5.0 equiv of **3a** was used.

Using the conditions described in our earlier report[58] on the defluorosilylation of aryl fluorides [Et$_3$SiBpin (1.5 equiv), Ni(COD)$_2$ (10 mol%), and potassium *tert*-butoxide (KOtBu, 2.5 equiv) in cyclohexane/tetrahydrofuran (THF) at room temperature], the expected biphenyl-phenylethyl-triethylsilane **4aa** was obtained regioselectively in 32% yield (entry 1, Table 1). Encouraged by this initial attempt, the optimum amount of base required was explored and 4.0 equiv of KOtBu was found to be the most suitable quantity (entries 1–3). We next varied the equivalents of Et$_3$SiBpin used and found that 2.0 equiv was the optimum amount (entry 4). After slight modifications to the solvent ratios, including testing a single solvent, the binary solvent cyclohexane/THF (8/1, v/v) was found to be most suitable (entry 5). Notably, reducing the Ni(COD)$_2$ catalyst loading to 1 mol% afforded **4aa** in 99% yield, which is similar to that obtained in entry 5, within 2.5 h without eroding the reaction efficiency (entry 6). Replacement of KOtBu with NaOtBu resulted in the formation of **4aa** in merely 15% yield (entry 7). In the absence of a base or when using other bases (LiOtBu, KOMe, or KHMDS), the desired product was not obtained (entries 8 and 9). Moreover, a large excess of **3a** had a negative effect on the reaction yield (entry 10). To ascertain the effect of Ni(COD)$_2$, we finally examined the reaction without Ni catalyst. We were very supersized that the transformation occurred efficiently even in the absence of Ni(COD)$_2$ to generate **4aa** in 94% yield (91% isolated yield, entry 11). The results are almost identical to the reaction with Ni catalysis (entry 6). We thus decided the further examination using the different substrates under the two type conditions, with or without Ni(COD)$_2$.

**Substrate scope**. With the optimized reaction conditions (Table 1, entries 6 and 11), we investigated the scope of the Ni-catalyzed or catalyst-free carbosilylation of the alkenes **3** with fluoroarenes **1** and Et$_3$SiBpin (Fig. 2). First, the scope of the aromatic fluorides was examined under catalyst-free conditions (entry 11, Table 1). A wide range of fluoroarenes that bear π-extended building blocks were efficiently converted into the corresponding defluorinative arylsilylation products (**4**) in good to high yield. For example, biphenyl products (**4aa**: 91%; **4ba**: 87%; **4ca**: 42%), a 1-naphthyl product (**4da**: 87%), and 4-(naphthalen-1-yl)phenyl (**4ea**: 90%) were all obtained using this methodology. Simple fluorobenzene

(**1f**) was also efficiently converted into arylsilylation product **4fa** (63%), whereas 4-methyl-substituted aryl fluoride **1g** provided a higher yield (**4ga**: 85%). The excellent chemoselectivity profile of this process is nicely illustrated by the tolerance of the reaction conditions toward functional groups such as ethers (-O-: **1h**, **1i**, and **1j**), CF$_3$ groups (**1k** and **1l**), or Cl groups (**1m**), all of which could potentially be cleaved with the C–F bond activation. The desired products **4** were obtained in the following yields: **4ha**: 75%, **4ia**: 73%, **4ja**: 69%, **4ka**: 50%, **4la**: 53%, **4ma**: 70%. It was most surprising to find that the CF$_3$ groups of **1k** and **1l** remained intact. We next repeated the same substrate scope by the best reaction conditions in the presence of Ni(COD)$_2$ (entry 6, Table 1). Although the yield products **4aa**–**na** were somewhat improved, the differences were not so significant, as shown in Fig. 2.

Encouraged by the fact that the transformation fundamentally does not require Ni catalyst, we continued the substrate scope under catalyst-free conditions (entry 11, Table 1). Next, we focused on the hetero-aromatic fluorides **1n**–**1u**. The nitrogen-containing hetero-aromatic fluorides **1n**–**1t** were successfully coupled with **3a** in the presence of Et$_3$SiBpin in good to high yield; a 1H-indole derivative (**4na**: 57%), 1H-pyrrole derivative (**4oa**: 32%), and a number of pyridine derivatives (**4pa**: 80%; **4qa**: 68%; **4ra**: 74%; **4sa**: 88%) were well-tolerated under the optimized conditions. Fluoro-indole (**1t**) and fluoro-benzofuran (**1u**), despite having several reactive aryl C($sp^2$)–H bonds, also participated well in this transformation, selectively furnishing carbosilylation products **4ta** (85%) and **4ua** (49%) via C–F bond cleavage, without the anticipated C–H activation–silylation reaction of the hetero-aromatic moiety. Notably, other silyl boronates such as PhMe$_2$SiBpin, $^n$Pr$_3$SiBPin, and $^t$BuMe$_2$SiBPin, could also be used in this transformation instead of Et$_3$SiBpin. The corresponding silylated products **4aa'**, **4aa"**, and **4aa"'** were obtained from fluoroarene **1a** and styrene (**3a**) in 47%, 88%, and 94% yields, respectively.

Next, we set out to evaluate the scope of the alkene component (**3**) of the reaction. A range of aryl alkenes were efficiently converted into the corresponding arylsilylation products **4** in the presence of aryl fluoride **1** and Et$_3$SiBpin. 2-Vinylnaphthalene **3b**, a simple polycyclic aromatic compound, successfully reacted with **1a** to afford the corresponding arylsilylation product (**4ab**, 58%). Styrene derivatives bearing Me (**3c**), $^t$Bu (**3d**), phenyl (**3e**), MeO

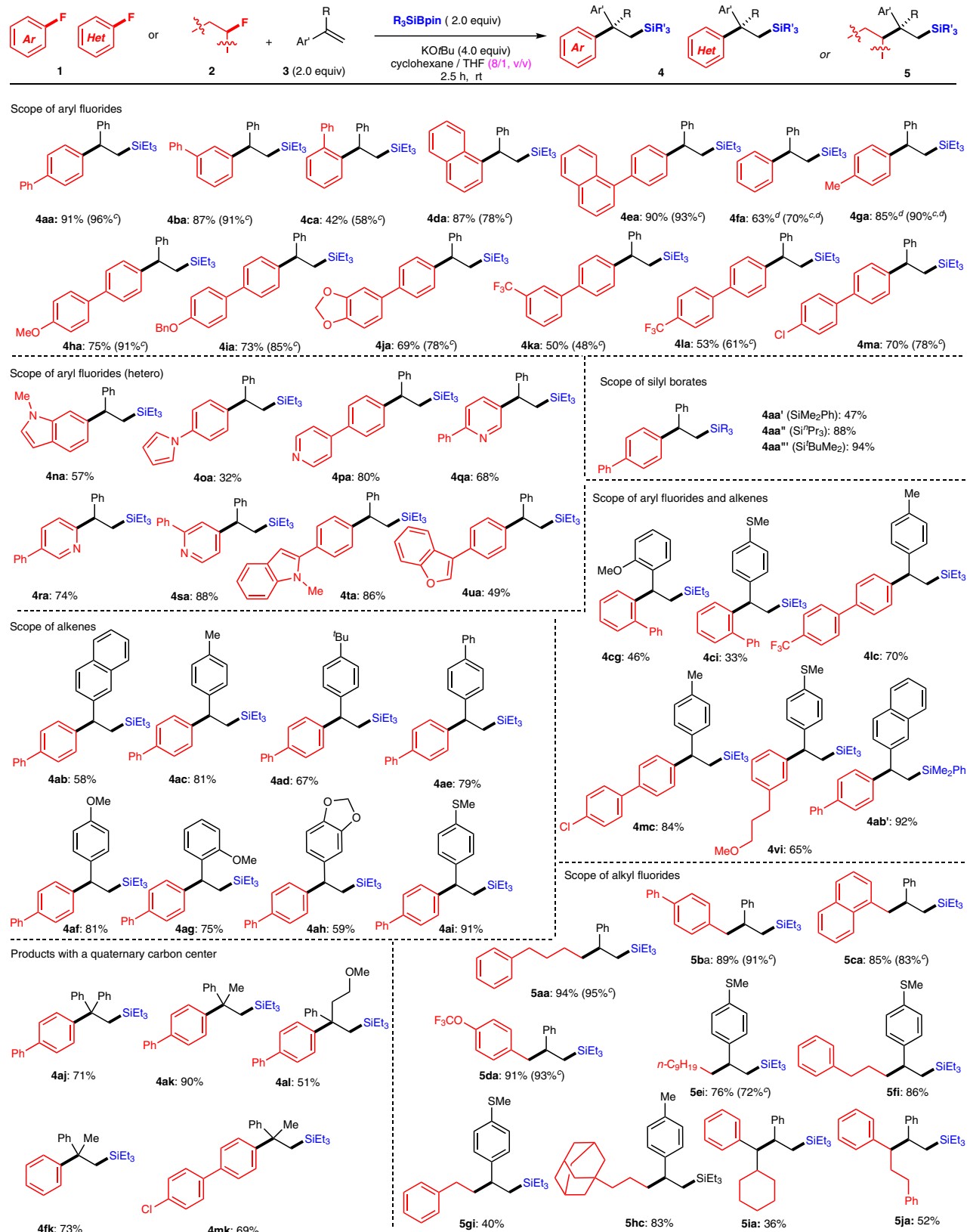

**Fig. 2 Carbosilylation of alkenes with silyl boronates and aryl/alkyl fluorides[a, b].** [a]The reaction was carried using **1** or **2** (0.2 mmol), **3** (2.0 equiv.), Et$_3$SiBpin, Me$_2$PhSiBpin, $^n$Pr$_3$SiBpin or $^t$BuMe$_2$SiBpin (2.0 equiv.), and KO$t$Bu (4.0 equiv.) in cyclohexane/THF (1.5 mL, 8/1, v/v) at room temperature. [b]Isolated yield is shown. [c]The reaction was carried out in the presence of Ni(COD)$_2$ (1.0 mol%). [d]0.4 mmol of **1** was used.

**Fig. 3 Substrates relevant to medicinal or materials chemistry. a** (±)-α-Tocopherol derivative. **b** Menthol derivative. **c, d** Estrone derivatives. **e** Liquid crystalline material derivative.

(3f, 3g), dioxole (3h), and MeS (3i) groups were all, even the highly sterically hindered *ortho*-MeO-styrene 3g, well-tolerated as substrates and furnished the desired products 4 in good to excellent yield (4ac: 81%; 4ad: 67%; 4ae: 79%; 4af: 81%; 4ag: 75%; 4ah: 59%, 4ai: 91%). It is important to note that this three-component coupling reaction tolerated any combination of the coupling components. A wide variety of substituted or non-substituted fluoroarenes 1, aryl alkenes 3, and silyl boronates $R_3SiBpin$ furnished the desired silylated alkanes 4 in good to high yield (4cg: 46%; 4ci: 33%; 4lc: 70%; 4mc: 84%; 4vi: 65%; 4ab': 92%). We also investigated the reaction of substituted styrene derivatives with a quaternary carbon center at the β-position of the silyl moiety. For example, α,α-diphenylethylene 3j, α-methylstyrene 3k, and (4-methoxybut-1-en-2-yl)benzene 3l reacted with fluoroarenes 1 under the standard reaction conditions to give the corresponding products 4 (4aj: 71%; 4ak: 90%; 4al: 51%; 4fk: 73%; 4mk: 69%). Furthermore, we attempted the defluorinative alkylsilylation of alkenes using alkyl fluorides with an inert $C(sp^3)$–F bond under the same reaction conditions. Primary alkyl fluorides 2a–2h efficiently reacted regioselectively with styrenes (3a, 3c, 3i) to afford the corresponding alkylsilylation products in up to 94% yield (5aa: 94%; 5ba: 89%; 5ca: 85%; 5da: 91%; 5ei: 76%; 5fi: 86%; 5gi: 40%; 5hc: 83%). Notably, the trifluoromethoxy ($CF_3O$) moiety of 2d remained intact to provide

5da. Secondary alkyl fluorides 2i and 2j also furnished the desired defluorinative alkylsilylation products 5ia and 5ja in 36% and 52% yield, respectively. The reaction using alkyl fluorides was also carried out in the presence of Ni(COD)$_2$ catalyst (1 mol%) under the best conditions (entry 6, Table 1). We obtained almost the same results (72–95%) as the yields without Ni catalyst. Thus, the defluorinative carbosilylation does not require Ni catalysis independent of the case of aryl or alkyl fluorides.

To highlight the synthetic utility of this three-component defluorinative carbosilylation reaction, we examined the functionalization of several drug derivatives with fluoroarene moieties (Fig. 3). (±)-α-Tocopherol derivative 1w successfully underwent carbosilylation to afford (±)-α-tocopherol derivative 4wa in 43% yield (Fig. 3a). (-)-Menthol-derived fluorobenzene 1x proceeded well under identical conditions to give carbosilylation product 4xa in 55% yield (Fig. 3b). Steroid derivatives 4ya and 4am were synthesized in 53% and 79% yield via (1) the defluorination of fluoro-incorporated estrone derivative 1y or (2) the defluorination of 1a with alkene- incorporated estrone derivative 3m (Fig. 3c, d). Moreover, the liquid crystalline material 1z was also successfully functionalized using this transformation with 3i to give 4zi in 41% yield (Fig. 3e).

Furthermore, we examined both the chemoselectivity and site-selectivity of the alkenyl moiety (Fig. 4). (E)-Buta-1,3-dien-1-

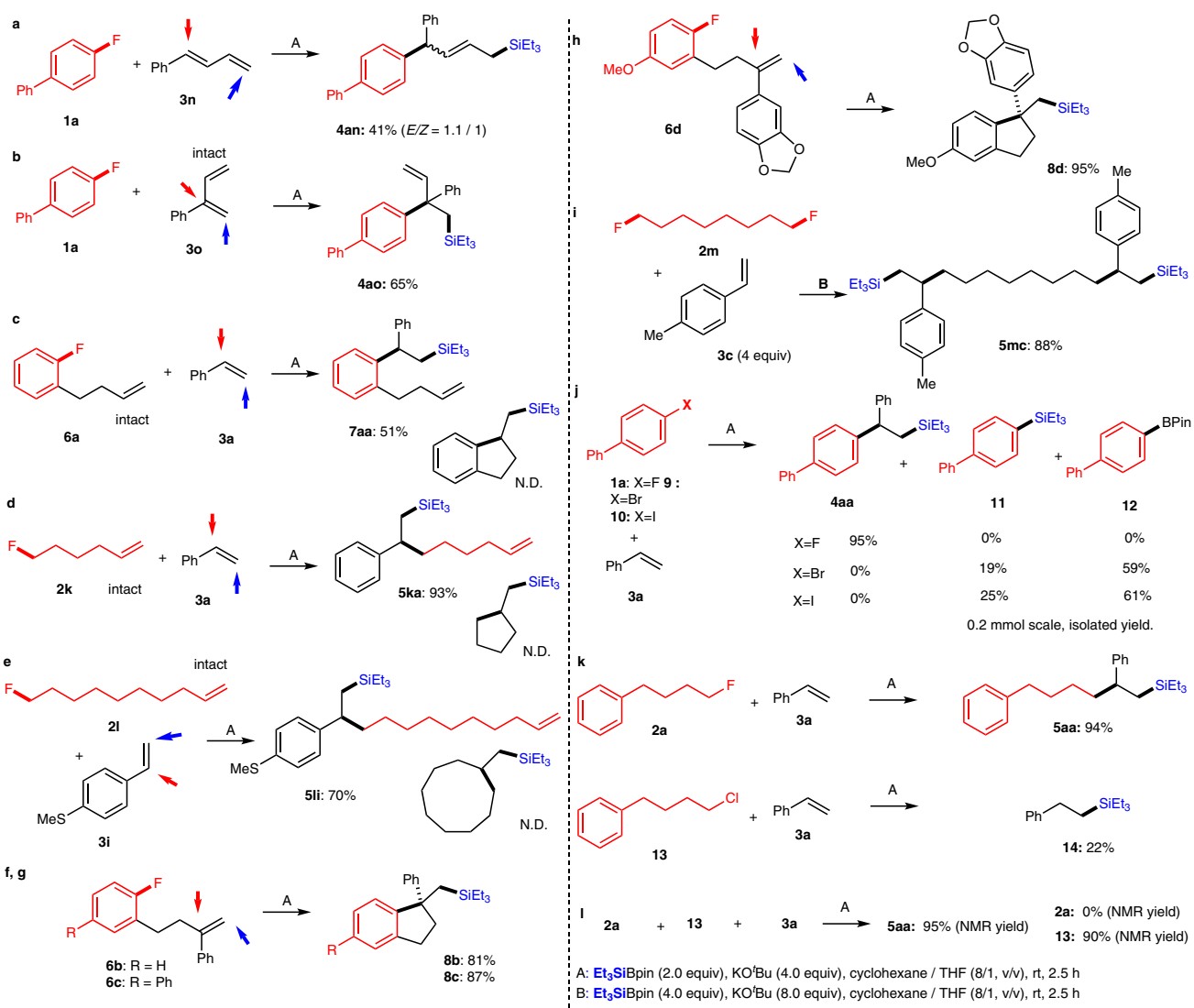

**Fig. 4 Chemo- and site-selective coupling reactions. a** Selective 1,4-type addition carbosilylation of 1,3-diene **3n**. **b** Selective 1,2-carbosilylation of 1,3-diene **3o**. **c–e** Selective three-component carbosilylation of styrenes (**3a** or **3i**) in the presence of an aliphatic alkene moiety in aryl fluoride **6a** or alkyl fluorides **2k**, or **2l**. **f–h** Intramolecular carbosilylation of **6b–d** to provide cyclized silylated products **8b–d**. **i** Five-component carbosilylation of styrene **3c** with 1,8-difluorooctane (**2m**). **j** Comparisons of aryl halides (**1a**, **9**, and **10**) for the carbosilylation. **k** Comparisons of alkyl halides (**2a** and **13**) for the carbosilylation. **l** The competitive reaction between **2a** and **13**.

ylbenzene (**3n**) was site-selectively converted into the 1,4-type arylsilylation product **4an** in 41% yield with an *E/Z* ratio of 1.1/1 (determined by [1]H nuclear magnetic resonance (NMR) analysis), whereas no 1,2-adduct was observed (Fig. 4a). Conversely, phenoprene **3o** preferably furnished **4ao** (65%), a compound with a quaternary carbon center, under the standard reaction conditions via the 1,2-arylsilylation process and not the 1,4-process (Fig. 4b). When we examined the reaction of 1-(but-3-en-1-yl)-2-fluorobenzene (**6a**) with **3a** in the presence of Et₃SiBpin, only the three-component condensation product **7aa** was obtained in 51% yield, whereas the intramolecular product was not detected (Fig. 4c). For the reaction of 6-fluorohex-1-ene (**2k**) and 10-fluorodec-1-ene (**2l**) having a terminal olefin moiety with styrenes **3a** and **3i**, the three-component condensation products **5ka** and **5li** were obtained in 93% and 70% yields, respectively, whereas the intramolecular products were not detected (Fig. 4d, e). In contrast, an intramolecular carbosilylation was achieved via the reaction of styrene-substituted fluoroarene **6b–6d** with Et₃SiBpin under identical conditions to furnish **8b–8d**, molecules with a quaternary carbon center, in 81–95% yield (Fig. 4f–h). The five-

component condensation was observed for the reaction of difunctionalized 1,8-difluorooctane (**2m**) with styrene **3c** and Et₃SiBpin to provide **5mc** in 88% yield (Fig. 4i). It should be noted that, under the applied conditions, bromo-, and iodo-substituted arenes **9**, **10** afforded a mixture of two-component condensation products of silylated **11** and borylated **12** compounds, whereas the targeted three two-component product **4aa** was not detected (Fig. 4j). We also attempted the reaction of alkyl chloride **13** instead of alkyl fluoride **3a** under the same conditions. Although the three-component product **5aa** was obtained by alkyl fluoride **3a**, the two-component product, triethyl(phenethyl)silane (**14**), made from **3a** and Et₃SiBPin, was obtained in 22% yield without incorporation of alkyl chloride **13** (Fig. 4k). Alkyl bromide and iodide were also not converted into **5aa** under identical conditions (Supplementary Fig. 5). Encouraged by the results, we attempted the chemoselective activation of the alkyl C–F bond over the alkyl C–Cl bond by the competitive reaction between **2a** and **13**. Interestingly, the alkyl fluoride **2a** was consumed to **5aa** (95%), whereas the alkyl chloride **13** was recovered (90%, Fig. 4l, more details in Supplementary Figs. 6–8). These behaviors show

**Fig. 5 Mechanistic study. a** Effect of TEMPO for carbosilylation of **1a**, **3a**, and Et₃SiBPin. **b, c** Radical clock experiments. **d** Effect of TEMPO for the reaction of alkyl fluoride **2k** and Et₃SiBPin. **e** Proposed reaction mechanisms.

the advantage of fluorine in our reaction system compared to commonly used halogens both in aromatic and aliphatic cases.

**Mechanistic study.** Based on this consideration and on the results obtained so far, a radical pathway seems to be a viable hypothesis. To gain an insight into the reaction mechanism, some experiments were undertaken. First the transformation of **1a** with **3a** to **4aa** under the best conditions was significantly inhibited by the addition of (2,2,6,6-tetramethylpiperidin-1-yl)oxyl (TEMPO) (Fig. 5a). These results strongly suggested the reaction involves radical species. We also attempted the same reaction in the presence of Ni catalyst. The same results were obtained. We next examined the radical clock experiment with the substrate containing a cyclopropyl moiety at the α-position of the styrene derivative **3p** (Fig. 5b). The silylated ring-opening product **15** was obtained as a major product in 55% yield and most of the aryl fluoride **1a** was recovered. On the other hand, the expected three-component product was not clearly observed, the trace of the desired material was detected only by gas chromatography–mass spectrometry (GCMS). The results are also in good agreement with the radical-mediated reaction mechanism, as the potential cyclopropyl carbinyl radical is known to spontaneously rearrange to allylcarbinyl radical. Furthermore, the substrate **6e** having both fluoroarene and styrene moieties was transformed into the silyl-cyclopropyl compound **16** in 76% yield (Fig. 5c), which also supports the radical pathway. The treatment of **2k** with Et₃SiBPin in the absence of styrenes **3** predominantly gave a silylalkene **17** in 73% yield via C–F bond activation, whereas the transformation was completely inhibited by the addition of TEMPO independent of the existence of Ni catalyst (Fig. 5d). Thus, the generation of radical species does not require the reaction with styrenes **3**.

Based on both the experiments' results here and in previous reports[44,67–74], we propose a single-electron transfer/radical-mediated carbosilylation reaction mechanism triggered by the known ability of KO*t*Bu to serve as a single-electron reductant[75,76] (Fig. 5e). First, Et₃SiBpin reacts with a molecule of KO*t*Bu to form an intermediate **A**. The formation of **A** has previously been

confirmed by the Avasare group based on density functional theory calculations[77]. We also confirmed the intermediate **A** by the ¹¹B-NMR and ²⁹Si-NMR study (Supplementary Figs. 12–15)[58]. Next, a single-electron transfer process would start by the additional amount of KO*t*Bu as a trigger[75,76]. Namely, a single electron is transferred from *t*-butoxy anion (⁻O*t*Bu) to the silicon atom of intermediate **A** to furnish a triethylsilyl radical ·SiEt₃ via a cleavage of the Si–B bond. The *t*-butoxy radical ·O*t*Bu is also generated, which would be captured by the borate anion **B**. The mechanisms for the cleavage of Si–B bonds are various; the radical-mediated Si–B bond cleavage[78–80] should be highly acceptable due to the experimental results using *t*BuOK[75,76]. The generated triethylsilyl radical ·SiEt₃ reacts with styrene **3a** to give a radical adduct **C**. The radical cascade process should happen between the radical species **C**, aryl **1** or alkyl fluorides **2**, *t*-butoxy radical ·O*t*Bu, and borate anion **B** in the transition state **TS-I**, where the C–F bond of aryl **1** or alkyl fluorides **2** is activated by the approach of K⁺. The boron atom in **B** would also participate in activation of the C–F bond. Finally, the C–C bond formation is completed under concomitant generation of stable **D** [BPin(O*t*Bu)₂]K (Supplementary Fig. 14)[58] and KF to furnish the desired three-component adduct **4** or **5**. The generation of benzyl radical species **C** was supported by the cyclopropyl experiments (Fig. 5b, c). It should be noted that through the experiments, we observed side products such as Et₃Si-SiEt₃ and double styrene adducts **18**. These formations can be explained by the dimerization of triethylsilyl radical ·SiEt₃ and the overreaction of benzyl radical **C** with styrene **3a**.

In conclusion, we have developed a carbosilylation of alkenes that uses silyl boronates and organic fluorides, and that proceeds via the activation of an inert C–F bond without a catalyst. This reaction should be initiated by the radical cleavage of Si–B bond via a single-electron transfer from *t*BuOK. A variety of β-functionalized silyl compounds can be synthesized efficiently and rapidly in good to excellent yield under very mild conditions at room temperature. The most significant feature of this protocol is its broad substrate scope. This highly efficient protocol accepts a variety of fluorides, including aryl and alkyl fluorides, and even

transforms sterically demanding secondary alkyl fluorides. A broad range of aryl alkenes, such as styrene derivatives and α-substituted aryl alkenes, are also tolerated by this method. Silyl boronates were identified as viable substrates. Moreover, the chemo- and site-selectivity displayed in this reaction are remarkable. Aryl alkenes selectively react in the presence of non-aryl alkenes and an intramolecular carbosilylation was achieved with substrates that possess both fluoroarene and aryl alkene moieties. The tolerance of the reaction toward different functional groups is also significant. Substrates with ether, CF₃, and hetero-aromatic moieties react smoothly without the detection of C–O cleavage, C–F bond activation, or C–H activation; only the inert C–F bonds of the fluoroarenes and fluoroalkanes are activated. Notably, this method also allows for the synthesis of silyl compounds with a quaternary carbon center at the β-position. The 1,4-type addition carbosilylation of 1,3-dienes was also achieved. Given the high number of fluorinated compounds that are commercially available, including structurally complex pharmaceuticals, agrochemicals, and functional materials, this protocol widens the potential utility of organosilicon compounds in organic synthesis, the structural design of lead drug compounds, and functional materials.

Finally, the limitations of the method should be mentioned. The conjugated diene is acceptable but electron-deficient acrylates and acrylamides are not suitable. Internal styrenes such as *cis*-stilbene and *trans*-β-methylstyrene did not react (Supplementary Figs. 1 and 2). The chemoselective activation of aromatic C–F bonds over aromatic C–Br and C–I bonds is difficult, whereas the only aromatic fluoride was transformed into the desired three-component product (Supplementary Figs. 3 and 4).

## Methods

**General procedure for the defluorinative carbosilylation of alkenes 3 using R₃SiBpin and aryl fluorides 1 or alkyl fluorides 2.** In a N₂-filled glovebox to a flame-dried screw-capped test tube were added organic fluorides **1** or **2** (0.20 mmol, 1.0 equiv), silyl boronates R₃SiBpin (0.4 mmol, 2.0 equiv), alkenes **3** (0.40 mmol, 2.0 equiv), KO*t*Bu (90 mg, 0.8 mmol, 4.0 equiv), and cyclohexane/THF (1.5 mL, 8/1, *v*/*v*) sequentially. The tube was then sealed and removed from the glovebox. The mixture was stirred at room temperature for 2.5 h. To the reaction tube was added hexane (5 mL) and then it was subjected to filter through a short silica pad, washed with Et₂O, and concentrated under vacuum, followed by 3-fluoropyridine (8.6 μL, 0.1 mmol) as an internal standard for NMR analysis. The mixture was then concentrated again to give the residue, which was purified by column chromatography on silica gel to give the corresponding carbosilylation products **4** or **5**.

## Data availability

The data supporting the findings of this study are available within the paper and its Supplementary Information. All relevant data are also available from the authors.

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

## Acknowledgements

This work was supported by JSPS KAKENHI grants JP 18H02553 and 21H01933 (KIBAN B, N.S.).

## Author contributions

N.S. conceived the concept. J.Z. optimized the reaction conditions. J.Z., B.J., Y.F., Z.Z, and T.I. surveyed the substrate scope. J.Z., B.J., and Y.F. prepared staring materials. N.S. directed the project. N.S. and J.Z. prepared the manuscript.

## Competing interests

The authors declare no competing interests.
