## [Peer Review File · Nature Communications]

Reviewers' Comments:

Reviewer #1:

Remarks to the Author:

This manuscript submitted by Shibata and co-workers describes the "Catalytic Carbosilylation of Alkenes using Silyl Boronates and Organic Fluorides via Selective C-F Bond Activation". This method allows rapid access to a wide variety of alkylsilanes from readily available alkenes, silylboranes, and aryl or alkyl fluorides by using Ni(COD)₂ as the catalyst. The scopes of alkenes, aryl or alkyl fluorides are reasonably broad, and it is remarkable that not only simple alkenes but also dienes are viable. A unique feature of the present work is selective activation of C-F bonds over C-O, C-S, and even C-Cl bonds in spite of thermodynamic stability of C-F bonds. This feature strengthens synthetic aspect of the present work because subsequent transformations of C-O, C-S, and C-Cl bonds of the products to prepare highly decorated alkylsilanes are possible. Thus, the reviewer recommends acceptance of this work in Nature Communications after minor revisions.

1) Regarding chemoselective Ar-F bond activation, what about chemoselective Ar-F bond activation over Ar-Br and Ar-I bonds? In this case, arylsilylation in the absence of Ni(COD)₂ as shown in entry 7 in Table 1 is a good choice.

2) Is it possible to achieve selective alkyl-F bond activation over alkyl-Cl bond in the alkylsilylation reaction?

3) There are several examples of alpha-substituted styrene derivatives. What happens when beta-substituted styrenes such as beta-methyl styrene is subjected to the standard conditions? Regardless of the results, it is better to add the result in the revised manuscript to show a limitation of the alkene scope.

Supporting Information

1) The purity of isolated products, 4qa, 5aa, and 5ga, does not seem enough. They should be repurified before publication.

2) The ChemDraw structure of 5ba in the ¹³C NMR spectrum is not correct.

Reviewer #2:

Remarks to the Author:

Shibata and co-workers reported the 1,2-carbosilylation of styrenes and (hetero)aryl alkenes derivatives by C-F activation of aryl and alkyl fluorides.

The difunctionalization of alkenes has emerged as an ideal strategy to generate molecular complexity by installing two (different) groups at the two terminus of a C=C bond. Many methods are now available to organic chemists to achieve such transformations and the array of suitable reagents is extremely wide and in continuous expansion. Among the carbon electrophiles, organic fluorides stand as very appealing precursors for alkene difunctionalization due to their abundance and the traditional inertness of C-F bonds. The method described by Shibata and co-workers inserts into this arena and elegantly showcases the possibility to use organic fluorides in combination with (hetero)aryl alkenes and Et₃SiBPin in a multicomponent fashion under very mild reaction conditions to prepare beta-disubstituted alkylsilanes.

The work is very well presented and the challenges clearly set. Nonetheless, the authors completely omitted to quote the existing methods for carbosilylation of alkenes although it is of essential importance because it gives the overview of the current state-of-the-art and which advancement the present work brings to the scientific community. For this reason, it is strongly suggested to insert the appropriate references in the manuscript and to add comments in this regards to the text.

The strategy described appears to be applicable to a great variety of substrates (up to 57 examples) and it tolerates a number of different functional groups. Particularly interesting is the

opportunity to extend such an approach to alkyl fluorides and to dienes (linear or branched) together with the chance to apply the transformation to an intramolecular setting, delivering cyclic alkyl silanes.

The reaction is claimed to proceed through a Ni-catalyzed C-F activation step followed by olefin migratory insertion into the Ni-Si bond and reductive elimination to release the product. The authors support this mechanistic hypothesis referring to previous works from the Martin's group and DFT calculations by the Avasare's groups, both regarding the silylation of aryl methyl ethers. Although some reaction components are similar, I would be rather careful when comparing these processes and sketch out any analogies.

Moreover, and maybe most importantly, the reaction described in the present paper proceeds to a considerable extent (49% conversion) without any nickel complex, suggesting it is not a Ni-catalyzed transformation or, at least, not the only important contribution. This aspect needs to be commented and clarified in a more exhaustive manner and if the authors do prove that also an actual Ni-catalyzed mechanism is operative, it is necessary to include the mechanism of a Ni-free process in Figure 5 in to account for the result in Table 1, entry 7.

Against the assumption of a Ni-catalyzed process, it needs to be considered that oxidative addition into a C(sp³)-F bond at room temperature seems very unlikely since the only reports about such a process require the use of stoichiometric amount of Grignard reagents. Considering the authors' proposal about this step, a 4-membered transition state with a Ni(0) interacting with the fluorine atom is envisioned. This transition state, however, likely lays very high in energy since it presents the coordination of an electron-rich Ni complex with a very electronegative F atom. Based on this consideration and on the results obtained by the authors, a radical pathway seems to be a viable hypothesis. Such a process might be triggered by the known ability of K⁺tBu⁻ to serve as single electron reductant.

However, the authors rule out the possibility of a radical pathway by the experiments reported in Figure 4c-d. These experiments might argue against a radical process involving the homolytic cleavage of the C-F bond, followed by radical addition to the pendant alkene moiety; in the same way they would be perfectly in line with the generation of a R³Si radical species which adds selectively to an electron-deficient olefin (polarity match) followed by a radical aromatic substitution mechanism (some examples of addition of C-radical to aryl group: Shi, *Angew. Chem. Int. Ed.* 2021, 60, 186–190; Liu, *Org. Lett.* 2015, 17, 2534–2537. For the addition of silyl radical to olefins: Alberti, Pedulli, *Reviews of Chemical Intermediates* 1987, 8, 207–246)

Simple radical clock experiments with cyclopropyl moieties on both the silyl fragment or at the α -position of the styrene derivative might easily confirm or disprove these hypothesis.

Although the method reported by the authors represents an important advancement to the realm of 1,2-difunctionalization of alkenes and C-F functionalization strategies, the report lacks of some results:

- In the scope of organic fluorides, only two purely alkyl fluoride are displayed (the ones leading to products 5aa and 5ei) whereas all the others are benzylic fluorides, which are notoriously easier to manipulate. It would be extremely important to show more examples of purely alkyl fluorides because it will enormously increase the value of the present method;
- In the scope of the silylating agents, only 2 different R³SiBPi_n reagents have been showed, importantly limiting the variety of silyl-groups which can be added across the C=C bond. It is suggested to show a greater diversity in this sense;
- The scope of olefins has been investigated extensively and the selectivity of the process towards electron-deficient alkenes has been clarified. Nonetheless, a very intuitive question arises: what is the outcome in presence of acrylates or other types of conjugated electron-deficient olefins? The results to this question should be added to set the potential and limitations of the current methodology.

In the manuscript, the authors compared their results to the ones reported by Hu in 2020 (ref. 31) stating that is "severely limited by the scope of the substrates tolerated under the applied reaction

conditions". Such statement is incorrect since the report from Hu disclosed a similar, if not wider, scope in terms of functional group compatibility thus it is suggested to rephrase the sentence accordingly.

All along the manuscript, the term Scheme is used in the text whereas the term Figure is used as footnote to the graphical part. It is necessary to address this point accordingly.

Another point to raise is the authors' claim of demonstrating the utility of the current methodology in the context of late-stage functionalization of drug derivatives. The molecules showed in Figure 3 do not present any additional challenge to the ones highlighted in the scope since they do not bring any particularly sensitive functional groups. Keeping this in mind, it will be more honest showing such derivatives in the same scheme of the scope without the need of overselling a method that is already innovative and practical enough for a publication in a high visible journal.

Taking in consideration the quality of the work and its novelty, together with all the points to be addressed/modified/added, it is suggested its acceptance in Nature Communication after major revisions.

Reviewer #3:

Remarks to the Author:

This Article by Shibata and coworkers describes a 1,2-carbosilylation of styrenyl alkenes using silyl boronates and aryl/alkyl fluorides with high chemoselectivity for cleavage of the C–F bond of the electrophilic coupling partner. Under optimized conditions, the reaction utilizes 1 mol% Ni(cod)₂ with 1 equivalents of organofluoride, 2 equivalents of alkene, 2 equivalents of silylboronate, and 4 equivalents of KOtBu (>50 examples, 33–96% yield). Importantly, as will be discussed below, the reaction proceeds in 49% yield in the absence of Ni(cod)₂. The reaction generally tolerates a wide range of functional groups in both the organofluoride and styrene scope. As mentioned above, the alkene scope is limited to conjugated alkenes, namely styrenes, which is the major limitation of this method. The methodology is used in a series of five applications involving the late-stage diversification of tocopherol, methol, estrone, and "liquid crystalline material derivative". These generally proceed in moderate to good yield (46–83%). Mechanistic experiments are generally lacking, but four experiments probing site-selectivity are carried out using styrene 1,3-dienes and ortho-alkenyl aryl fluorides as cyclization probes. For a linear 1,3-dienes a formal 1,4-carbosilylation is achieved in 42% yield, and for a branched 1,3-dienes the styrenyl alkene undergoes 1,2-carbosilylation in 63% yield. For cyclization probes, an intramolecular 1,2-carbosilylation is observed for an aryl fluoride with a tethered distal styrene fragment (88% yield), and there is observed selectivity for styrenyl alkenes over aliphatic alkenes. A rough sketch of a catalytic cycle is proposed at the end of the manuscript that draws heavily from uncited DFT work and prior work. Overall, the mechanism is not fully supported by any mechanistic experiments conducted in the manuscript itself, and other potential pathways that are perhaps more likely are not considered or discussed.

The Supporting Information is well-written overall and contains sufficient quality.

In general, the field of transition-metal-catalyzed intermolecular alkene 1,2-difunctionalization has gained rapid interest in recent years, benefiting from extensive studies. This is particularly true of those involving styrenes. While it is true that aryl fluorides are scarcely used as electrophilic fragments in these types of three-component couplings, the general field of C–F cleavage using silyl cooperativity and/or transition metal catalysis has also benefited from a wide breadth of research, especially in the last decade. Several citations are absent from the authors on this front and must be included, or at least a review if citation limits are exceeded (i.e. J. Am. Chem. Soc. 2015, 137, 39, 12470–12473; Angew. Chem. Int. Ed. 2018, 57, 9103–9107; Review: Chem. Rev. 2017, 117, 13, 8710–8753). Aside from these issues regarding acknowledgment of prior art, there are some aspects of the chemistry that merit revision and/or discussion. As mentioned above, the authors note that in the absence of transition metal the reaction proceeds in 49% yield, which is a substantial amount of product. Given that some product yields are in a range close to the control reaction, it begs the question of how this process is impacted by addition of a nickel catalyst. In

the Supporting Information, the authors provide a selection of other transition metals with 3.5 equivalents of base, but in this case even Ni(cod)₂ gives 38% yield compared to 30% without transition metal. This table must be conducted under optimized conditions as it seems that based on this data set there can be no conclusion made about the nature, or requirement, of the nickel catalyst. It is also a bit strange that an extra 0.5 equivalent of base triples the reaction yield. The section labeled "Mechanistic Study" contains almost no experimental evidence for the proposed cycle that is drawn, and the experiments that precede this section are not discussed in the context of what insights can be drawn from the selectivity achieved. This significantly diminishes the quality of the manuscript, especially in the realm of nickel catalysis in which mechanistic insight is invaluable given that the overall reaction is prototypical in 1,2-difunctionalization. It is unacceptable to state that DFT conducted by the Avastare group "confirms" a mechanism without any citation. This is underscored further by the fact that the proposed mechanism is not supported by any experiments in the present manuscript. While I acknowledge that if the reaction is in fact completely nickel-mediated, it is likely to occur via similar mechanisms to the authors' prior work in defluorosilylation, these types of experiments are still merited. Other concerns with the catalytic cycle will be described below, but the major curiosity is the observed reactivity without catalyst. While there are some elements of synthetic utility in the overall reaction, my assessment is that the present state of the manuscript is not currently at the appropriate level for publication in Nature Communications. For this reason, I recommend that this manuscript is heavily revised according to concerns above and the following specific comments seriously considered prior to any publication:

- (1) As stated above, Ref. 26 alone is not sufficient to capture the validity of the field of C–F bond functionalization. Other citations must be added and the sentence adjusted to properly reflect the prior literature.
- (2) While it is true that carbosilylation is not as common as other reaction modes, several citations are again absent. Again, other citations must be added to properly reflect the field, such as: *J. Am. Chem. Soc.* 1995, 117,9814 –9821; *Org. Lett.* 2011, 13,1828 –1831; *J. Org. Chem.* 2016, 81,3065 –3069; *Angew. Chem. Int. Ed.* 2019, 58, 17068 –17073.
- (3) In general, all transition metals probed in SI should be conducted at optimized conditions. This will provide better landscape for studying potential mechanisms.
- (4) In Table 1, the footnote says 0.2 mmol of 3b, but there is no 3b in the table.
- (5) Does the reaction tolerate internal styrenes? This would provide valuable insight on the nature of the proposed insertion mechanism (syn vs anti) and expand the utility greatly. Do acrylates, enamides, or other similarly activated alkenes function in this reaction? This could add another layer of selectivity that would be particularly enabling, or expand the scope.
- (6) The 1,3-diene studies are particularly interesting given the difference in selectivity for the two isomers. Is this purely a result of internal styrenes being inaccessible substrates or does it have to do with stability of a *n*-allyl/benzyl species? This can be screened with and without nickel.
- (7) In general, the results with alkyl fluorides are promising and very useful. For cyclization studies, how does an alkyl fluoride behave? For example, 6-fluoro-hexene and the analogous styrene would be nice probes.
- (8) As a general question, is borylation ever observed using these reagents? What are the side products in these reactions?
- (9) For the proposed mechanism, the relevant citations must be provided and drawn out more clearly. For example, why does the Ni-center have two η -2 COD ligands instead of a single η -4 COD ligand that would be more stable? Is there concrete evidence for this structural intermediate? To truly understand the role of the nickel catalyst, especially since the reaction proceeds quite well without it, I suggest that reaction progress kinetic analysis (RPKA) be conducted with Ni(cod)₂. It is possible that there are two distinct processes occurring, one of which is not metal-mediated, and this could also be interesting. I would imagine a few scenarios in which either the rate of nickel

catalysis is orders of magnitude faster than the background, in which the overall process is solely derived from a proposed catalytic cycle involving nickel, or both the background reactivity and nickel-catalyzed process are additive. In any case, this type of mechanistic interrogation would greatly enhance the scholarship of the manuscript.

RESPONSE TO REVIEWERS:

Reviewer #1 (Remarks to the Author):

This manuscript submitted by Shibata and co-workers describes the “Catalytic Carbosilylation of Alkenes using Silyl Boronates and Organic Fluorides via Selective C-F Bond Activation”. This method allows rapid access to a wide variety of alkylsilanes from readily available alkenes, silylboranes, and aryl or alkyl fluorides by using Ni(COD)₂ as the catalyst. The scopes of alkenes, aryl or alkyl fluorides are reasonably broad, and it is remarkable that not only simple alkenes but also dienes are viable. A unique feature of the present work is selective activation of C-F bonds over C-O, C-S, and even C-Cl bonds in spite of thermodynamic stability of C-F bonds. This feature strengthens synthetic aspect of the present work because subsequent transformations of C-O, C-S, and C-Cl bonds of the products to prepare highly decorated alkylsilanes are possible. Thus, the reviewer recommends acceptance of this work in Nature Communications after minor revisions.

Answer: Thank you for your high evaluation.

1) Regarding chemoselective Ar-F bond activation, what about chemoselective Ar-F bond activation over Ar-Br and Ar-I bonds? In this case, arylsilylation in the absence of Ni(COD)₂, as shown in entry 7 in Table 1 is a good choice.

Answer: Thank you for your question. The chemoselective of Ar-F bond activation over Ar-Br and Ar-I bonds is very good. No arylsilylation was observed for Ar-Br and Ar-I; instead, a mixture of aryl boronate and aryl silane products were formed. These results are now in the main text of the revised manuscript.

X	Yield of 4aa (%)	Yield of By-P-1	Yield of By-P-2
F	95	0	0
Br	0	19	59
I	0	25	61

0.2 mmol scale, Yield shown were isolated

2) Is it possible to achieve selective alkyl-F bond activation over alkyl-Cl bond in the alkylsilylation reaction?

Answer: The chemoselective alkyl-F bond activation over the alkyl-Cl bond is perfect. Alkyl chlorides did not give the three-component products at all. These results are now in the revised manuscript's main text.

3) There are several examples of alpha-substituted styrene derivatives. What happens when beta-substituted styrenes such as beta-methyl styrene is subjected to the standard conditions? Regardless of the results, it is better to add the result in the revised manuscript to show a limitation of the alkene scope.

Answer: Thank you for your comments. The *cis*-stilbene and *trans*-β-methylstyrene did not give the desired products. We now commented it on the revised manuscript as “Limitations of the Study”.

Supporting Information

1) The purity of isolated products, 4qa, 5aa, and 5ga, does not seem enough. They should be repurified before publication.

Answer: Revised.

2) The ChemDraw structure of 5ba in the 13C NMR spectrum is not correct.

Answer: Revised.

Reviewer #2 (Remarks to the Author):

Shibata and co-workers reported the 1,2-carbosilylation of styrenes and (hetero)aryl alkenes derivatives by C-F activation of aryl and alkyl fluorides. The difunctionalization of alkenes has emerged as an ideal strategy to generate molecular complexity by installing two (different) groups at the two terminus of a C=C bond. Many methods are now available to organic chemists to achieve such transformations and the array of suitable reagents is extremely wide and in continuous expansion. Among the carbon electrophiles, organic fluorides stand as very appealing precursors for alkene difunctionalization due to their abundance and the traditional inertness of C-F bonds. The method described by Shibata and co-workers inserts into this arena and elegantly showcases the possibility to use organic fluorides in combination with (hetero)aryl alkenes and Et₃SiBPin in a multicomponent fashion under very mild reaction conditions to prepare beta-disubstituted alkylsilanes.

The work is very well presented and the challenges clearly set.

Answer: Thank you for your kind evaluation.

Nonetheless, the authors completely omitted to quote the existing methods for carbosilylation of alkenes although it is of essential importance because it gives the overview of the current state-of-the-art and which advancement the present work brings to the scientific community. For this reason, it is strongly suggested to insert the appropriate references in the manuscript and to add comments in this regards to the text

Answer: Thanks for your suggestion. We commented in the introduction part, and references were revised.

The strategy described appears to be applicable to a great variety of substrates (up to 57 examples) and it tolerates a number of different functional groups. Particularly interesting is the opportunity to extend such an approach to alkyl fluorides and to dienes (linear or branched) together with the chance to apply the transformation to an intramolecular setting, delivering cyclic alkyl silanes. The reaction is claimed to proceed through a Ni-catalyzed C-F activation step followed by olefin migratory insertion into the Ni-Si bond and reductive elimination to release the product. The authors support this mechanistic hypothesis referring to previous works from the Martin's group and

DFT calculations by the Avasare's groups, both regarding the silylation of aryl methyl ethers. Although some reaction components are similar, I would be rather careful when comparing these processes and sketch out any analogies. Moreover, and maybe most importantly, the reaction described in the present paper proceeds to a considerable extent (49% conversion) without any nickel complex, suggesting it is not a Ni-catalyzed transformation or, at least, not the only important contribution. This aspect needs to be commented and clarified in a more exhaustive manner and if the authors do prove that also an actual Ni-catalyzed mechanism is operative, it is necessary to include the mechanism of a Ni-free process in Figure 5 in to account for the result in Table 1, entry 7.

Answer: Thank you very much for your critical matter. Indeed, we noticed that the reaction proceeds well without Ni (49%). We thus carefully repeated the reaction again under the best conditions with or without Ni. We now noticed that a similarly high yield of the arylsilylation product has resulted even without Ni. This finding suggested that Ni(cod)₂ is not necessary. The early result of the 49% in the initial study could be something technical matter, purity, or activity of the reagent Et₃SiBpin performed. We thus re-examined thoroughly ALL the reactions without Ni-catalyst very carefully. All the yields without Ni are fundamental as similar to with Ni catalyst in the original manuscript, while the Ni-conditions are slight better (around 5%). Thus, the experimental results were entirely re-examined and revised. A new reaction mechanism was proposed. I have sincerely acknowledged your comments.

Against the assumption of a Ni-catalyzed process, it needs to be considered that oxidative addition into a C(sp³)-F bond at room temperature seems very unlikely since the only reports about such a process require the use of stoichiometric amount of Grignard reagents. Considering the authors' proposal about this step, a 4-membered transition state with a Ni(0) interacting with the fluorine atom is envisioned. This transition state, however, likely lays very high in energy since it presents the coordination of an electron-rich Ni complex with a very electronegative F atom. Based on this consideration and on the results obtained by the authors, a radical pathway seems to be a viable hypothesis. Such a process might be triggered by the known ability of KOtBu to serve as single electron reductant.

Answer: Thank you for your suggestion. We agree with you, and all the reaction proceeds well without Ni. Your advice of a radical pathway should be very, very reasonable. We have examined several more reactions to ascertain the radical mechanism accordingly. Thank you very much once again.

However, the authors rule out the possibility of a radical pathway by the experiments reported in Figure 4c-d. These experiments might argue against a radical process involving the homolytic cleavage of the C-F bond, followed by radical addition to the pendant alkene moiety; in the same way they would be perfectly in line with the generation of a R₃Si radical species which adds selectively to an electron-deficient olefin (polarity match) followed by a radical aromatic substitution mechanism (some examples of addition of C-radical to aryl group: Shi, *Angew. Chem. Int. Ed.* 2021, 60, 186–190; Liu, *Org. Lett.* 2015, 17, 2534–2537. For the addition of silyl radical to olefins: Alberti, Pedulli, *Reviews of Chemical Intermediates* 1987, 8, 207–246). Simple radical clock experiments with cyclopropyl moieties on both the silyl fragment or at the α-position of the styrene derivative might easily confirm or disprove these hypothesis.

Answer: Thank you again and again for your great suggestion. We attempted the radical clock experiments with cyclopropyl moieties. We also examined reactions in the presence of TEMPO. Your advice is perfect, and the results strongly support the radical mechanism.

Although the method reported by the authors represents an important advancement to the realm of 1,2-difunctionalization of alkenes and C-F functionalization strategies, the report lacks of some results:

- In the scope of organic fluorides, only two purely alkyl fluorides are displayed (the ones leading to products 5aa and 5ei) whereas all the others are benzylic fluorides, which are notoriously easier to manipulate. It would be extremely important to show more examples of purely alkyl fluorides because it will enormously increase the value of the present method;

Answer: We now expanded the examples of purely alkyl fluorides from 2 to 10 examples (in Figures 2 and 4).

- In the scope of the silylating agents, only 2 different R₃SiBPin reagents have been showed, importantly limiting the variety of silyl-groups which can be added across the C=C bond. It is suggested to show a greater diversity in this sense;

Answer: Yes, we used two different R₃SiBPin reagents, i.e., Et₃SiBPin and PhMe₂SiBPin, in the original manuscript. Now we added two more examples using Pr₃Si-BPin and TBDMS-BPin in the revised manuscript (in Figure 2).

- The scope of olefins has been investigated extensively and the selectivity of the process towards electron-deficient alkenes has been clarified. Nonetheless, a very intuitive question arises: what is the outcome in presence of acrylates or other types of conjugated electron-deficient olefins? The results to this question should be added to set the potential and limitations of the current methodology.

Answer: We attempted the reaction using *tert*-butyl 2-phenylacrylate, benzyl acrylate, and *N*-methyl-*N*-phenylacrylamide under the standard conditions with or without Ni, we detected no desired products. Thus, this should be the limitation. We now commented on the revised manuscript, as "Limitations of the Study".

In the manuscript, the authors compared their results to the ones reported by Hu in 2020 (ref. 31) stating that is “severly limited by the scope of the substrates tolerated under the applied reaction conditions”. Such statement is incorrect since the report from Hu disclosed a similar, if not wider, scope in terms of functional group compatibility thus it is suggested to rephrase the sentence accordingly.

Answer: Thank you for your suggestion, we rephrased the sentence accordingly.

All along the manuscript, the term Scheme is used in the text whereas the term Figure is used as footnote to the graphical part. It is necessary to address this point accordingly.

Answer: Thank you for your suggestion, we revised.

Another point to raise is the authors’ claim of demonstrating the utility of the current methodology in the context of late-stage functionalization of drug derivatives. The molecules showed in Figure 3 do not present any additional challenge to the ones highlighted in the scope since they do not bring any particularly sensitive functional groups. Keeping this in mind, it will be more honest showing such derivatives in the same scheme of the scope without the need of overselling a method that is already innovative and practical enough for a publication in a high visible journal.

Answer: Thanks for your suggestion. We agree and deleted the word “late-stage” and mentioned “late-stage functionalization of drug derivatives.”

Taking in consideration the quality of the work and its novelty, together with all the points to be addressed/modified/added, it is suggested its acceptance in Nature Communication after major revisions.

Answer: Thanks for your kind evaluation.

Reviewer #3 (Remarks to the Author):

This Article by Shibata and coworkers describes a 1,2-carbosilylation of styrenyl alkenes using silyl boronates and aryl/alkyl fluorides with high chemoselectivity for cleavage of the C–F bond of the electrophilic coupling partner. Under optimized conditions, the reaction utilizes 1 mol% Ni(cod)₂ with 1 equivalents of organofluoride, 2 equivalents of alkene, 2 equivalents of silylboronate, and 4 equivalents of KOtBu (>50 examples, 33–96% yield). Importantly, as will be discussed below, the reaction proceeds in 49% yield in the absence of Ni(cod)₂. The reaction generally tolerates a wide range of functional groups in both the organofluoride and styrene scope. As mentioned above, the alkene scope is limited to conjugated alkenes, namely styrenes, which is the major limitation of this method. The methodology is used in a series of five applications involving the late-stage diversification of tocopherol, methol, estrone, and “liquid crystalline material derivative”. These generally proceed in moderate to good yield (46–83%). Mechanistic experiments are generally lacking, but four experiments probing site-selectivity are carried out using styrene 1,3-dienes and ortho-alkenyl aryl fluorides as cyclization probes. For a linear 1,3-dienes a formal 1,4-carbosilylation is achieved in 42% yield, and for a branched 1,3-dienes the styrenyl alkene undergoes 1,2-carbosilylation in 63% yield. For cyclization probes, an intramolecular 1,2-carbosilylation is observed

for an aryl fluoride with a tethered distal styrene fragment (88% yield), and there is observed selectivity for styrenyl alkenes over aliphatic alkenes. A rough sketch of a catalytic cycle is proposed at the end of the manuscript that draws heavily from uncited DFT work and prior work. Overall, the mechanism is not fully supported by any mechanistic experiments conducted in the manuscript itself, and other potential pathways that are perhaps more likely are not considered or discussed.

Answer: Thank you for your comments. These are the same comments from Reviewer 2. We thoroughly re-examined ALL the reaction with and without Ni, and we re-discovered Ni is not necessary for this transformation. We revised all the results and reaction mechanisms based on the revised results accordingly.

The Supporting Information is well-written overall and contains sufficient quality.

Answer: Thank you.

In general, the field of transition-metal-catalyzed intermolecular alkene 1,2-difunctionalization has gained rapid interest in recent years, benefiting from extensive studies. This is particularly true of those involving styrenes. While it is true that aryl fluorides are scarcely used as electrophilic fragments in these types of three-component couplings, the general field of C–F cleavage using silyl cooperativity and/or transition metal catalysis has also benefited from a wide breadth of research, especially in the last decade. Several citations are absent from the authors on this front and must be included, or at least a review if citation limits are exceeded (i.e. J. Am. Chem. Soc. 2015, 137, 39, 12470–12473; Angew. Chem. Int. Ed. 2018, 57, 9103–9107; Review: Chem. Rev. 2017, 117, 13, 8710–8753).

Answer: Thank you. We revised the citations.

Aside from these issues regarding acknowledgment of prior art, there are some aspects of the chemistry that merit revision and/or discussion. As mentioned above, the authors note that in the absence of transition metal the reaction proceeds in 49% yield, which is a substantial amount of product. Given that some product yields are in a range close to the control reaction, it begs the question of how this process is impacted by addition of a nickel catalyst. In the Supporting Information, the authors provide a selection of other transition metals with 3.5 equivalents of base, but in this case even Ni(cod)₂ gives 38% yield compared to 30% without transition metal. This table must be conducted under optimized conditions as it seems that based on this data set there can be no conclusion made about the nature, or requirement, of the nickel catalyst. It is also a bit strange that an extra 0.5 equivalent of base triples the reaction yield (**Answer: solvent system and equiv of Et₃SiBpin are different. Entries 2, 3 4 and 5, Table 1**). The section labeled “Mechanistic Study” contains almost no experimental evidence for the proposed cycle that is drawn, and the experiments that precede this section are not discussed in the context of what insights can be drawn from the selectivity achieved. This significantly diminishes the quality of the manuscript, especially in the realm of nickel catalysis in which mechanistic insight is invaluable given that the overall reaction is prototypical in 1,2-difunctionalization. It is unacceptable to state that DFT conducted by the Avasare group “confirms” a mechanism without any citation. This is underscored further by the fact that the proposed mechanism is not supported by any experiments in the present manuscript. While I acknowledge that if the reaction is in fact completely nickel-mediated, it is likely to occur via similar mechanisms to the authors’ prior work in defluorosilylation, these types of experiments are still merited. Other concerns with the catalytic cycle will be described below, but the major curiosity is the observed reactivity without catalyst. While there are some elements of synthetic utility in the overall reaction, my assessment is that the present state of the manuscript is not currently at the appropriate level for publication in Nature Communications. For this reason, I

recommend that this manuscript is heavily revised according to concerns above and the following specific comments seriously considered prior to any publication:

Answer: Thank you for your comments. These are the same comments from Reviewer 2. We thoroughly re-examined the reaction with and without Ni, and we re-discovered Ni is not necessary for this transformation. We revised all the results and reaction mechanisms based on the revised results accordingly.

(1) As stated above, Ref. 26 alone is not sufficient to capture the validity of the field of C–F bond functionalization. Other citations must be added and the sentence adjusted to properly reflect the prior literature.

Answer: Thank you for your comments. We revised the sentences and citations accordingly.

(2) While it is true that carbosilylation is not as common as other reaction modes, several citations are again absent. Again, other citations must be added to properly reflect the field, such as: J. Am. Chem. Soc. 1995, 117,9814–9821; Org. Lett. 2011, 13,1828–1831; J. Org. Chem. 2016, 81,3065–3069; Angew. Chem. Int. Ed. 2019, 58, 17068–17073.

Answer: Thank you for your comments. We revised the sentences and citations accordingly.

(3) In general, all transition metals probed in SI should be conducted at optimized conditions. This will provide better landscape for studying potential mechanisms.

Answer: Thank you for your comments. After careful re-examination, we re-found that the reaction does not require nether Ni nor transition metal catalysts, and we revised the manuscript accordingly.

(4) In Table 1, the footnote says 0.2 mmol of 3b, but there is no 3b in the table.

Answer: Thank you for your comments. We revised.

(5) Does the reaction tolerate internal styrenes? This would provide valuable insight on the nature of the proposed insertion mechanism (syn vs anti) and expand the utility greatly. Do acrylates, enamides, or other similarly activated alkenes function in this reaction? This could add another layer of selectivity that would be particularly enabling, or expand the scope.

Answer: This is the same question as reviewer #1 and reviewer #2. They are the limitation of this method. We now commented on the revised manuscript.

(6) The 1,3-diene studies are particularly interesting given the difference in selectivity for the two isomers. Is this purely a result of internal styrenes being inaccessible substrates or does it have to do with stability of a π -allyl/benzyl species? This can be screened with and without nickel.

Answer: Thanks for your question. This is purely a result of internal styrenes being inaccessible substrates. We re-examined the reaction without Ni and confirmed it again. The internal styrenes are the limitation, and it is commented on in the revised manuscript. as “Limitations of the Study”.

(7) In general, the results with alkyl fluorides are promising and very useful. For cyclization studies, how does an alkyl fluoride behave? For example, 6-fluoro-hexene and the analogous styrene would be nice probes.

Answer: Thank you for your suggestion. We attempted the reactions using 6-fluoro-hexene with or without styrene under the standard conditions (now, without Ni). Desired carbosilylation product, triethyl(2-phenyloct-7-en-1-yl)silane, was formed in 93% yield in the presence of styrene. Without styrene, triethyl(hex-5-en-1-yl)silane was detected, while no cyclized product was formed. These results indicated that simple alkenes are unreactive. They

are now commented on in the revised manuscript.

(8) As a general question, is borylation ever observed using these reagents? What are the side products in these reactions?

Answer: Thanks for your question. We didn't detect the borylation products. The detectable side products are listed below. We commented it in the revised manuscript.

(9) For the proposed mechanism, the relevant citations must be provided and drawn out more clearly. For example, why does the Ni-center have two eta-2 COD ligands instead of a single eta-4 COD ligand that would be more stable? Is there concrete evidence for this structural intermediate? To truly understand the role of the nickel catalyst, especially since the reaction proceeds quite well without it, I suggest that reaction progress kinetic analysis (RPKA) be conducted with Ni(cod)₂. It is possible that there are two distinct processes occurring, one of which is not metal-mediated, and this could also be interesting. I would imagine a few scenarios in which either the rate of nickel catalysis is orders of magnitude faster than the background, in which the overall process is solely derived from a proposed catalytic cycle involving nickel, or both the background reactivity and nickel-catalyzed process are additive. In any case, this type of mechanistic interrogation would greatly enhance the scholarship of the manuscript.

Answer: Thanks for your comments. Since we re-found the reaction does not require Ni, we revised the reaction mechanism, including the radical process.

Reviewers' Comments:

Reviewer #1:

Remarks to the Author:

The revisions on my questions 1 and 2 are not suitable. Other revisions are appropriate.

The results in Figure 4j do not show a chemoselective activation of the C-F bond because bromobenzene and iodobenzene are converted to 11 and 12. Chemoselective activation means that C-F bond activation occurs while C-Br and C-I bonds are intact. The results of 4-bromofluorobenzene and 3-iodofluorobenzene in Figure S1 indicate that chemoselective activation of C-F bonds over C-Br and C-I bonds is not possible.

The results in Figure 4k indicate possibility of chemoselective activation of C-F bond of alkylfluorides over C-Cl bond of alkylchlorides. Competitive reaction between 2a and 13 would be much better to show the chemoselectivity clearly.

Reviewer #2:

Remarks to the Author:

Shibata and co-workers reported on the catalyst-free 1,2-carbosilylation of styrenes and (hetero)aryl alkenes derivatives by C-F activation of aryl and alkyl fluorides.

The difunctionalization of alkenes has emerged as an ideal strategy to generate molecular complexity by installing two (different) groups at the two termini of a C=C bond. Many methods are now available to organic chemists to achieve such transformations and the array of suitable reagents is extremely wide and in continuous expansion.

Among the carbon electrophiles, organic fluorides stand as very appealing precursors for alkene difunctionalization due to their abundance and the traditional inertness of C-F bonds.

The method described by Shibata and co-workers inserts into this arena and elegantly showcases the possibility to use organic fluorides in combination with (hetero)aryl alkenes and R₃SiBPin in a multicomponent fashion under very mild reaction conditions to prepare of beta-disubstituted alkylsilanes.

The work is very well presented and the challenges clearly set. The strategy is applicable to a great variety of substrates (up to 68 examples) and it tolerates a number of different functional groups. Particularly interesting is the opportunity to extend such an approach to alkyl fluorides and to dienes (linear or branched) together with the chance to apply the transformation to an intramolecular setting, delivering cyclic alkyl silanes.

The authors took in consideration the comments and suggestions provided by the Referees and run further mechanistic experiments, which appear to discard the former proposal of a Ni-catalyzed process in favour of a transition metal-free radical reaction.

Moreover, additional examples of purely alkyl fluorides and of silylating agents have been showed, considerably widening the scope of the present methodology.

Taking in consideration the quality of the work, its novelty and the additional investigation carried out, it is suggested its acceptance in Nature Communication.

Reviewer #3:

Remarks to the Author:

In its newly revised state, the manuscript meets the standards for publication in Nature Communications. My prior concerns have been mostly addressed. This being stated, I recommend the newly revised version be published after the following minor revisions:

(1) Figure 2: It seems that Ni(cod)₂ remains on the reaction arrow, despite it being denoted as only being used in cases of superscript "C". I believe this should be removed from above the reaction arrow.

(2) Figure 5a: "TEMPO" is spelled incorrectly above the reaction arrow.

(3) For the confirmation of substrates by ¹¹B NMR, it would be good to include the confirmed result in the Supporting Information. This is generally useful since it is an interesting point of discussion, and broadly applies to Figure 5. For example, if I understand correctly, the DFT methodology is not present in this paper (cited appropriately), but it would be clearer to indicate which data in Figure 5 are from the present report and from prior publications in the figure itself. This could also be commented on more clearly in the Supporting Information with a crisp description and appropriate citation for the reader.

Thank you very much for your kind decision. We are now happy to send the revised manuscript. The revisions have been made according to the comments/suggestions by reviewers. The revisions are indicated with yellow highlights in the revised manuscript. Our point-by-point responses to the reviewers' requests are shown below. Since all the comments are clearly responded, I hope the revised manuscript should be suitable for the acceptance.

Reviewer #1 (Remarks to the Author):

The results in Figure 4j do not show a chemoselective activation of the C-F bond because bromobenzene and iodobenzene are converted to 11 and 12. Chemoselective activation means that C-F bond activation occurs while C-Br and C-I bonds are intact. The results of 4-bromofluorobenzene and 3-iodofluorobenzene in Figure S1 indicate that chemoselective activation of C-F bonds over C-Br and C-I bonds is not possible.

Answer: Thank you for your comments. Your comments are correct that the chemoselective activation of C-F bonds over C-Br and C-I bonds on aromatic system is difficult. I attempted two more reactions to show the difficulty of chemoselective activation of aryl C-F bonds over C-Br, and C-I bonds. The results are now shown in Figures S3 and S4 and commented it in the section of "Limitations of the Study", as "The chemoselective activation of aromatic C-F bonds over aromatic C-Br and C-I bonds is difficult, while the only aromatic fluoride is transformed into the desired three-component product (Figures S3 and S4)."

R	Yield of 4
1f: R = H	63% (isolated)
R = Br	4%
R = I	5%

0.2 mmol scale, NMR yield shown.

The results in Figure 4k indicate possibility of chemoselective activation of C-F bond of alkylfluorides over C-Cl bond of alkylchlorides. Competitive reaction between 2a and 13 would be much better to show the chemoselectivity clearly.

Answer: Thank you for your comments. Based on your suggestion, we did the competitive reaction between 2a and 13 under the best conditions. Excitingly, the chemoselective activation of alkyl C-F bond over C-Cl bond was observed. The results are now shown in Figures 4k and 4l with text, and more details are in Figures S5-8 in SI.

X	Yield of 5aa
F (2a)	96%
Cl (13)	0%
Br (13-Br)	0%
I (13-I)	0%

0.2 mmol scale, NMR yield shown.

Reviewer #2 (Remarks to the Author):

Shibata and co-workers reported on the catalyst-free 1,2-carbosilylation of styrenes and (hetero)aryl alkenes derivatives by C-F activation of aryl and alkyl fluorides.

The difunctionalization of alkenes has emerged as an ideal strategy to generate molecular complexity by installing two (different) groups at the two termini of a C=C bond. Many methods are now available to organic chemists to achieve such transformations and the array of suitable reagents is extremely wide and in continuous expansion.

Among the carbon electrophiles, organic fluorides stand as very appealing precursors for alkene difunctionalization due to their abundance and the traditional inertness of C-F bonds.

The method described by Shibata and co-workers inserts into this arena and elegantly showcases the possibility to use organic fluorides in combination with (hetero)aryl alkenes and R₃SiBPin in a multicomponent fashion under very mild reaction conditions to prepare of beta-disubstituted alkylsilanes.

The work is very well presented and the challenges clearly set. The strategy is applicable to a great variety of substrates (up to 68 examples) and it tolerates a number of different functional groups. Particularly interesting is the opportunity to extend such an approach to alkyl fluorides and to dienes (linear or branched) together with the chance to apply the transformation to an intramolecular setting, delivering cyclic alkyl silanes.

The authors took in consideration the comments and suggestions provided by the Referees and run further mechanistic experiments, which appear to discard the former proposal of a Ni-catalyzed process in favour of a transition metal-free radical reaction. Moreover, additional examples of purely alkyl fluorides and of silylating agents have

been showed, considerably widening the scope of the present methodology. Taking in consideration the quality of the work, its novelty and the additional investigation carried out, it is suggested its acceptance in Nature Communication.

Answer: Thank you very much for your high evaluation. Based on your previous suggestions, the quality of the manuscript has been much improved. Thank you once again.

Reviewer #3 (Remarks to the Author):

In its newly revised state, the manuscript meets the standards for publication in Nature Communications. My prior concerns have been mostly addressed. This being stated, I recommend the newly revised version be published after the following minor revisions:

(1) Figure 2: It seems that Ni(cod)₂ remains on the reaction arrow, despite it being denoted as only being used in cases of superscript "C". I believe this should be removed from above the reaction arrow.

Answer: Thank you for your suggestion. It was revised.

(2) Figure 5a: "TEMPO" is spelled incorrectly above the reaction arrow.

Answer: Thank you for your suggestion. It was revised.

(3) For the confirmation of substrates by ¹¹B NMR, it would be good to include the confirmed result in the Supporting Information. This is generally useful since it is an interesting point of discussion, and broadly applies to Figure 5. For example, if I understand correctly, the DFT methodology is not present in this paper (cited appropriately), but it would be clearer to indicate which data in Figure 5 are from the present report and from prior publications in the figure itself. This could also be commented on more clearly in the Supporting Information with a crisp description and appropriate citation for the reader.

Answer: Thank you for your suggestion. The formation **A** from Et₃SiBpin and tBuOK is confirmed by reported DFT (reference 77). The formation **A** was also assigned by ¹¹B NMR (reference 58). Now, we indicated the references into the Figure 5e. We also have newly taken ¹¹B NMR, and ²⁹Si NMR to confirm the substrates **A** as well as **D**. These new NMR data and their references were also added into Figure 5e and more details and copies of NMRs are shown in SI (Figures S12-15).

e)

Reviewers' Comments:

Reviewer #1:

Remarks to the Author:

The revisions on this round addressed all of my concerns. The quality of the manuscript meets the standard of Nature Communications. The manuscript is ready to be accepted.

Reviewer #3:

Remarks to the Author:

The newly revised document addresses all prior concerns and is suitable for publication in Nature Communications.